# A Multidimensional Analysis of Released COVID-19 Location-Based Mobile Applications

**Theodoros Oikonomidis \*, Konstantinos Fouskas** 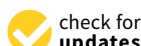 **and Maro Vlachopoulou**

ISeB Lab., Department of Applied Informatics, University of Macedonia, Egnatia 156, 546 36 Thessaloniki, Greece;
kfouskas@uom.edu.gr (K.F.); mavla@uom.edu.gr (M.V.)
\* Correspondence: t.oikonomidis@uom.edu.gr

**Abstract:** The spread of coronavirus disease (COVID-19) has triggered a series of responses worldwide ranging from traveling restrictions and shelter-in-place orders to lockdowns, contact tracing, social distancing, and other mitigation measures. To assist with contact tracing and ensure the safety of citizens, a significant number of mobile applications has been developed, utilizing geospatial information and proximity sensing. We perform a thorough research on seven digital databases (Appbrain, e-Health Hub, GDPRhub, "fs0c131y", News Sites, Appstore, and Google Play), identifying a total of 160 apps regarding COVID-19 related to our research questions. The aim of this research is to identify the main categories of apps and analyze their functions based on a proposed framework of by mapping aspects that affect their functionalities regarding Services, Technology, Societal & Business, and Legal aspects. As the world comes to the new normal, the utilization of these apps might become more essential for more mobile users and developers. The new encryption protocols that are established are also in favor of this argument. Future work can utilize our framework to further examine the development, design, and adoption of such mobile applications.

**Keywords:** COVID-19 location-based mobile apps; location-based technologies; digital contact tracing; data privacy; coronavirus disease



## 1. Introduction

The coronavirus (COVID-19) pandemic has triggered a series of issues on citizens' activities and has promoted social distancing on most of our activities. As previous pandemics and natural disasters have been faced by governments with the assistance of digital technology, the new coronavirus has challenged again our response efficacy. Location-based mobile applications (LBMAs) related to COVID-19 have been launched providing functionalities that can assist in limiting COVID-19 spread. These include home monitoring surveillance, social monitoring, and contact tracing of apps' users. The pandemic of COVID-19 required rapid and effective action, leading to increased use by governments and/or private companies of mobile applications installed on smartphones to fight the spread of the virus. These measures and mobile apps adopted by public and private authorities during the health crisis uphold the protection of individuals with regard to the processing of personal data. Privacy rights and data protection have a pivotal role in building and maintaining trust in mobile solutions, ensuring that the use of such mobile applications takes into full consideration of human dignity and integrity. As COVID-19 tracing is a highly sensitive area of research, a wide range of publications address potential problems in terms of technology, security, privacy, and governance of their development and use. Although most countries developed apps to aid proximity and contact tracing, some other countries invested efforts in apps aimed at fulfilling other purposes, such as dissemination of information to citizens (news, general instructions to avoid infections, geographical maps), medical support (self-diagnosis, reporting, information to access to health services), mandatory and non-mandatory applications related to quarantine

enforcement, forms for movement during lockdowns, map travel patterns, contact and proximity tracing, and monitoring rule violations.

A report entitled "Digital solutions to fight COVID-19" was published in October 2020 by the Data Protection Unit of the Council of Europe listing the digital solutions (including mobile apps) adopted or planned in the context of the COVID-19 in 55 countries from Africa, Europe, and Latin America (Parties to Convention 108) in order to give insights on the legal and technical measures adopted and their impact on the rights to privacy and data protection. The report identifies a number of shortcomings in some of the legal and technical measures adopted and calls on governments to ensure the transparency of digital solutions, in order to ensure greater respect for privacy and data protection rights. It also notes that despite numerous calls for coordination and interoperability of digital solutions to control the spread of the COVID-19 pandemic, countries have implemented individual applications, thus limiting the efficiency and comparability of the measures taken [1]. Specifically, the research of Cho et al. (2020) ensures that privacy is a central feature of conversations surrounding mobile contact tracing apps and encourages community efforts to develop alternative effective solutions with stronger privacy protection for the users [2]. Vinuesa et al. (2020) found that mobile applications for contact tracing differ in technology and methods and are concerned about their implications for privacy and human rights. They propose a socio-technical framework to evaluate current approaches and concerns related to the development, deployment, and usage of tracing apps, illustrating its implementation with three applications [3]. Additionally, Behne et al. (2021) investigated the research problem of digital solutions to overcome the pandemic, more closely examining the limited effectiveness and scope of the governmental COVID-19 tracing apps, using the German COVID-19 tracing app (Corona-Warn-App) as an example. Practical relevant findings can be transferred to other COVID-19 tracing apps as well as to other crisis management applications derived from interdisciplinary knowledge [4].

Since smartphone apps related to COVID-19 continue to emerge and evolve and vary on included functions, Ramakrishnan et al. (2020) provided a comprehensive and adaptable framework to help individuals assess the growing number of COVID-19 mobile apps. The ultimate aim is to give guidance for the future development and provision of these apps [5]. Furthermore, Kondylakis et al. (2020) conducted a systematic review to shed light on studies found in the scientific literature that have used and evaluated mobile apps for the prevention, management, treatment, or follow-up of COVID-19. They examined the bibliographic databases Global Literature on Coronavirus Disease, PubMed, and Scopus to identify papers focusing on mobile apps for COVID-19 showing evidence of their real-life use. According to their findings, mobile apps have been found to benefit citizens, health professionals, and decision-makers in facing the pandemic crisis [6].

The aim of this paper is the identification and mapping of a significant sample of well-known LBMAs worldwide that have been launched in response to the Coronavirus outbreak, presenting the different categories of services and functionalities provided to governments, health authorities/ World Health Organization, and citizens. We try to expand the previous work of theoretical frameworks developed to address digital contact tracing issues by identifying the main categories of apps and analyzing their functions. The use of digital technologies must be carried out strictly in line with human rights by centralized or decentralized authorities. Our research analyzes and evaluates the applications, understanding their evolution and usage. This allows us to assess the impact of LBMAs on both helping and raising concerns about the extended usage of this technology and its effectiveness [3].

The remainder of this paper is structured as follows. The next section is the theoretical background, initiating the usage of a COVID-19 LBMAs framework. Section 3 outlines the methodology adopted, where we describe the search selection criteria, and screening process for the identification of eligible LBMAs for inclusion. This is followed by presenting the framework of mapping LBMAs by examining aspects that affect their functionalities in the services, technology, societal and business, and legal aspects, and present relevant

alternative features of the 160 LBMAs examined in the results section. The paper concludes with a discussion on all the aspect categories that derive from the use of COVID-19 LBMAs regarding inhibitory aspects such as governmental and privacy issues. We also present their limitations and propose specific directions for future research in the conclusion section.

## 2. Theoretical Background

COVID-19 LBMAs have been developed in response to the coronavirus outbreak and serve as a digital assistant tool for smartphone users. The continuous change of type on LBMAs indicates the lack of knowledge regarding the usage and effectiveness of these apps. The categorization of those apps is not clear which might derive from the lack of evaluation of the proposed COVID-19 LBMAs.

Research that examines the service aspects of COVID-19 apps has not provided a categorization between the apps based on their functionalities. The result is that the majority of research refers to COVID-19 LBMAs as "contact tracing apps" when studies with in-depth analysis suggest additional categories based on the different functionalities they perform. For example, recent research [7,8] does not use a specific term when referring to COVID-19 LBMAs, when relevant studies [2] are using the term "contact tracing" which does not include or cover the different categories and functionalities among them.

COVID-19 LBMAs utilize specific technologies that help them perform COVID-19 related location services. On the technology aspects research identifies the smartphone antennas and chips that are utilized by COVID-19 LBMAs to provide their offering services [9]. Additionally, these aspects are correlated with societal and business aspects in relevant scientific publications, and privacy concerns are examined based on the authority that manages the function of these COVID-19 LBMAs [10].

Another major issue regarding COVID-19 LBMAs is whether they are monitored by the private sector (business) or a governmental authority something that influences the utilization of the users' data, how they are treated. Regarding Societal and Business aspects, there are studies that provide additional information based on the developers of COVID-19 LBMAs as the main entity that has the responsibility of ethical use of the apps [11].

Ongoing research also studies the legal aspects that influence the functionalities and services that can be utilized by mobile users. The data that are also gathered by the developers are also a part of this category of aspects which usually are utilized for research purposes, but in many countries, they are an additional digital tool to monitor the spread of COVID-19 [12].

There is an opportunity for mobile health to create a platform that is easily accessible to everyone so that the public has access to useful public-health-related information [13]. The recent technology advancement benefits the growth of e-health services where publicly available information has been proved to be extremely useful in the field of healthcare [14]. Within the last months, we witnessed a rapid change in major activities related to both private and business life. The spread of COVID-19 has raised a series of responses in terms of reorganization of work and personal activities, such as, among others, the rise of remote working, the cancellation or digitization of large-scale events, the limitation of citizens' movements, the closure of shops and the rise of e-commerce, and many more. Information technology has become an aid in crisis response, as it has been in previous cases such as Severe Respiratory Syndrome (SARS) and Asian Tsunami disasters [15]. A significant type of aid to this type of crisis, as indicated in relevant research by Boulos et al. (2019), is the use of geographic information systems (GIS) and methods, which have been widely utilized and have significant potential in better responses—also in the case of COVID-19 [16]. Some of the above can be the basis for future development of COVID-19 LBMAs, where the already launched COVID-19 LBMAs can be transformed for different needs of the mobile health aid industry.

There are a limited number of studies on COVID-19 LBMAs that record all applications involved and categorize them or evaluate their effectiveness depending on the geographical area in which they are applied. Based on previous research [5] our study focuses on

covering the previous research gaps by providing a different approach on categorization features and evaluation criteria of COVID-19 LBMAs [6]. Therefore, we propose a research framework (Figure 1) that organizes all the apps based on four types of aspects: Service, technology, societal and business, and legal aspects. According to this framework, the evaluation and differentiation between the existing LBMA COVID-19 can be performed by examining their effectiveness and usage, based on aspects related to service functionalities, technological and safety issues, privacy, human rights and governmental concerns, and societal and business guidelines.

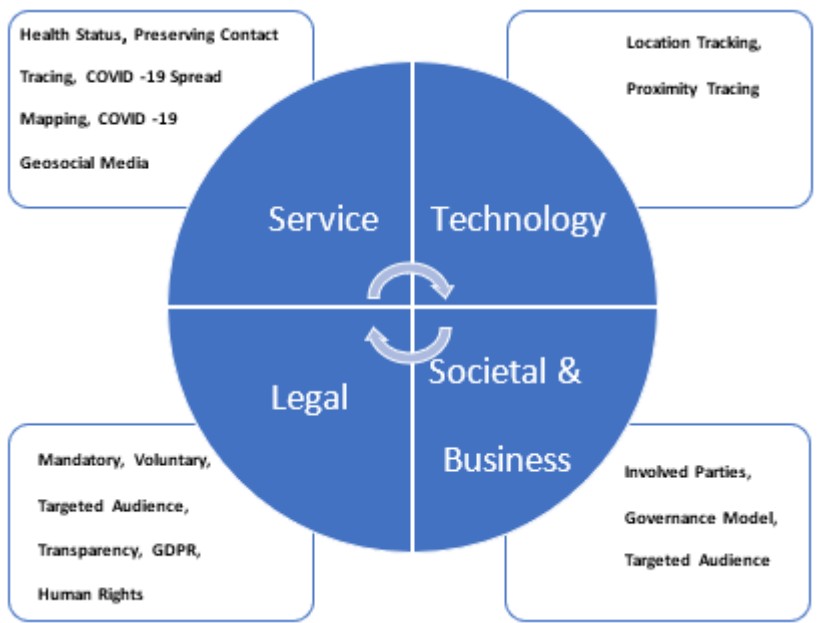

**Figure 1.** COVID-19 LBMAs Framework.

The research questions that are addressed by this research are the following:

(1) "In what ways and how LBMAs can help to better target and design measures to contain and slow the spread of the COVID-19 pandemic?" (Type of usage, services, and functionalities)
(2) "Which technologies have been used?" (Utilized technologies and possible combinations of them in order to respect individual's privacy)
(3) "Which parties are involved in their governance, which is the targeted audience and do their serve multiple purposes at once?" (Involved parties, governance model, targeted audience, and development of multipurpose LBMAs)
(4) Is the use of mobile applications mandatory or voluntary, and what other legal issues are raised?" (Mandatory or voluntary usage, respect of privacy, data collection compliant to the General Data Protection Regulation)

## 3. Materials and Methods

Coronavirus tracking and mapping is a key approach for LBMAs. A significant number of COVID-19 LBMAs from various international contexts have been collected and categorized taking into consideration a multidimensional analysis related to services, technology, societal and business, and legal aspects. According to our research design regarding seven online application databases (namely Appbrain, e-Health Hub, GDPRhub, Fs0c131y, News Sites, Appstore, Google Play), our study was conducted during the period March to June 2020 including worldwide relevant results and was updated in November 2020. Specifically, utilizing seven different research databases, we performed a systematic analysis by using a series of keywords such as "Covid19," "Coronavirus," "Corona," and "COVID-19" [17]. We accessed results worldwide by utilizing Virtual Private Network

(VPN) to overcome geographical restrictions. The full names of the mobile applications (found in Appendix A) were in some cases translated from the original language into English so that they can be presented accordingly in Appendix A. Following three-phase selection criteria (depicted in Figure 2) and expert analysis, the mobile apps relevant to our research were reduced to 160. 79 of these apps were categorized as Health Status apps, 46 as Preserving Contact Tracing, 23 as COVID-19 Spread Mapping, 4 as COVID-19 Geosocial Media, and 8 as multipurpose apps [18].

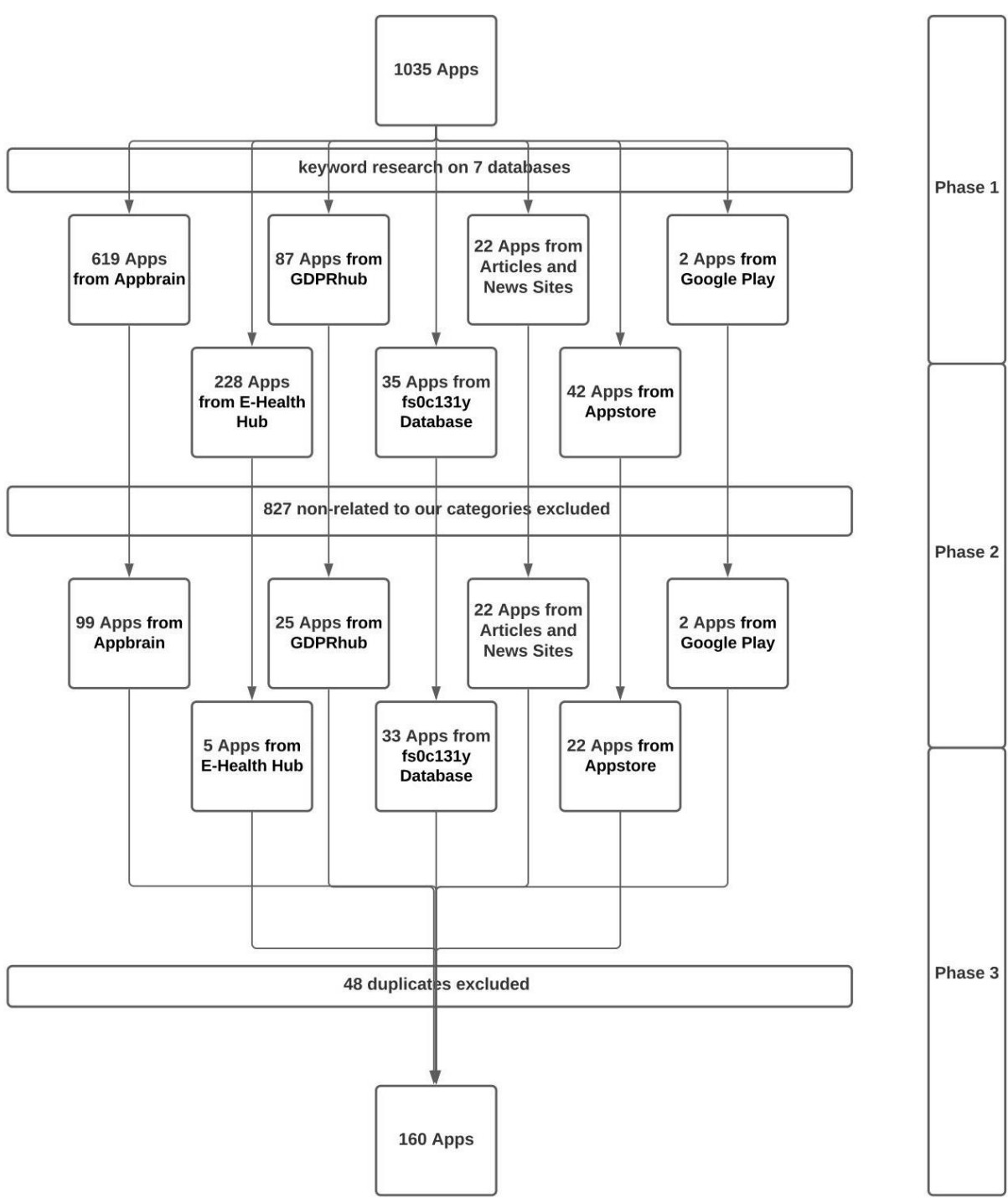

**Figure 2.** COVID-19 LBMAs selection.

In the first phase, we applied relevant keywords and identified 1035 apps in total. Those 619 Apps were from Appbrain Digital Database, 228 Apps from E-Health Hub, 87 Apps from GDPRhub, 35 Apps from "fs0c131y" Database, 22 from articles and News Sites 42 from Appstore, and 2 from Google Play. In the second phase of analysis, we excluded 827 apps that were not relevant to our research scope. Specifically, there were commercial apps that focused on e-business growth (such as apps assisting in grocery or indoor fitness apps) that were eventually excluded on this round. These applications were not explicitly developed for combating COVID-19 or are relevant to the usage of location-based or proximity tracing technologies. All mobile apps remaining were selected on the criteria to be able to assist digitally and control the users' exposure to the virus or identify potential infection exposure to the COVID-19. This round concluded by the examination of all selected applications by two independent external experts.

The next round included 99 apps from Appbrain, 5 Apps from E-Health Hub, 25 from GDPRhub, 33 from "fs0c131y" Database, 22 from Articles and News Sites, 22 from Appstore, and 2 from Google Play. In the third phase of the analysis, we excluded 48 duplicate mobile applications that were identified in different databases, which led us to the final sample of 160 COVID-19 LBMAs. For the final sample of 160 apps, based on predefined recording protocols, we performed a secondary examination on the different databases to identify information that would be relevant to our research. These included the publisher of the application, service aspects, utilized technologies, governance model, targeted users, voluntary/mandatory type of usage, and the platform that was utilized, among others.

## 4. The Framework of Mapping and Evaluating COVID-19 LBMAs

In the following paragraphs, we analyze the four categories of the proposed framework for COVID-19 LBMAs. We also provide additional information on the number of apps that belong to each category in the results section.

### 4.1. Service Aspects of COVID-19 LBMAs

Focusing on the type of functionalities/services provided by all included in our re-search LBMAs and taking into consideration previous literature attempts, from a user-centric value chain perspective, we suggest the following four categories of COVID-19 LBMAs "Health Status, Preserving Contact Tracing, COVID-19 Spread Mapping, and COVID-19 Geosocial Media" apps [17,19–21]. The functionalities of all included mobile apps were screened individually by three external researchers to con-firm the functionality/service categorization of all included mobile apps, reading the summary and explanation given by the developers for each application in each respective database. Any disagreement was discussed until consensus was achieved.

Health Status apps provide e-health services related to the COVID-19 including the provision ability to perform remote self-diagnosis, mostly based on the symptom's recognition for the user's perspective and distance treatment and online consultation with a health authority [22]. A typical example of Health Status is the "Covid Symptom Tracker" (app no 18) that asks questions to the mobile user for a preliminary check regarding COVID-19. If the users have been found suspicious for COVID-19 infection it is suggested that they get tested in the nearest hospital clinic for COVID-19.

The Preserving Contact Tracing function is utilized in case someone infected has reached an unsafe distance with another LBMA user, such as in the case of "Mahakavach" (app no 106), where users contribute such information to detect potentially infected people from Coronavirus. This category also includes quarantine or home monitoring for people infected with the virus, and in most cases, must be quarantined for at least 14 days [23]. For instance, "StayHomeSafe" (app no 116) traces the borders of the user's home and sends a notification in case someone accidentally leaves their premises. An additional function that falls into this category is route tracking, which is justified as means of finding places visited by infected people to avoid infecting healthy people ("Hamagen" app no 102). A

further service provided by applications in this category is safe distance using GPS (global position system) and other technologies, such as Bluetooth, to locate a person who may have reached an unsafe distance from the end-user of the app ("MyTrace" app no 107) [24].

COVID-19 spread mapping involves all the LBMAs that include live maps of COVID-19 information regarding the spread of the disease. A typical example of this category is "Corona Map" (app no 128) where users can be informed of the COVID-19 case number in each geographical area of Saudi Arabia.

COVID-19 geosocial media apps include COVID-19 communication and related content regarding COVID-19. For example, users of "uLouder" (app no 150) instantly join a local communication platform in their region. The app enables users to assist older people who need help in this emergency, such as assistance for their groceries since their exposure to the crowd might increase their risk of infection. Furthermore, the "WeChat" (app no 151) has raised awareness during COVID-19 since it triggered companies and developers around the world to cooperate in providing pandemic-specific services [25].

### 4.2. Technology Aspects of COVID-19 LBMAs

#### 4.2.1. Location Tracking Technologies

The importance of location has been highlighted by previous researchers that refer to the location as "actionable information" that can be used to improve human health [16]. They are utilized from LBMAs to estimate the exact location of the users. Cellular Network is a commonly utilized signal to track the exact location of mobile users. Network carriers can track the location of the phone user even if it is not a smartphone. Cellular data cannot be shared with anyone outside of the carrier company or the mobile application provider unless it is a case of an emergency where the authorities are asking for additional information. It utilizes the carrier signal strength in order to estimate the spatial location of the smart device user [26]. The main method for location estimation through Cellular Network is triangulation, where the different distances between the smartphone and the cellular towers consist of a triangle where the user is identified in terms of location. The accuracy of this method is from hundreds of methods up to kilometers. This accuracy is achieved by calculating the distance of the antennas of each mobile phone company with the user's mobile phone.

The GPS of smartphones offers location services on the majority of the LBMAs by utilizing the satellite signal to estimate the position of the user on the earth [16]. The GPS antenna needs four satellites to provide geospatial information to the user with an approximate accuracy of 4.9 m [27].

Location tracking technologies are centralized technologies that collect users' information for governmental and research purposes. Specifically, when the first pandemic wave arrived, the U.S. Government inspected the Cellular Network data to watch the places of high human traffic that consisted of potential spreading regions of COVID-19, and further measures were taken such as restrictions and shutdowns [28].

#### 4.2.2. Proximity Tracing Technologies

Bluetooth consists of proximity technology that is a simplified term for Bluetooth Low Energy and is useful for indoor and outdoor proximity services. It usually functions in conjunction with other smart devices where they can estimate their differential distance, usually with the utilization of an LBMA [29]. The accuracy of proximity information is usually around 5 m [30]. Mainly the received signal strength identification (RSSI) is a measurement of the power present in a received radio signal and utilized to estimate the relative distance between different smart devices. As a complementary technology to Bluetooth, contact-tracing tokens have been released that identify automatically close contacts of the user without the use of any mobile application. The advantage of this method is increased privacy for two main reasons. Data is available only to approved authorities of each country, and it is impossible for a software update to surreptitiously turn on location data or other sensors without notifying the user [31].

Another proximity tracing technology is the Wi-Fi antenna that utilizes the signal for Internet access on smart devices. The Wi-Fi positioning system can be implemented utilizing multiple points of access and by using trilateration upon an end-user that is utilizing the received Wi-Fi signal on a smartphone. The research in this area is focused on increasing the emitting or the receiving signal on smart devices and finding the optimal solutions for accurate positioning when it comes to indoor environments [32].

### 4.2.3. Tracing Protocols

Decentralized privacy-preserving proximity tracing (DP-3T) is a protocol created for Bluetooth technology that ensures that individuals' exchanged data remain anonymous. Its purpose is to create a protocol that will help in reducing the spread of COVID-19 by notifying people with digital contact tracing of potential exposure to the virus. This feature aims to break the chain of transmission of COVID-19 [3]. The European Union (EU) created the Pan-European Privacy-Preserving Proximity Tracing (PEPP-PT/PEPP) that enables the utilization of LBMA by keeping the private information of end-users safe. It was created specifically for the COVID-19 pandemic inside Europe as an alternative to the decentralized privacy-preserving proximity tracing (DP-3T) protocol, and both are linked to the utilization of Bluetooth. This protocol is considered centralized, which may compromise the privacy of the smart device user. However, it cannot prevent the user from using a pseudonym instead of their real name. In case of an emergency, such as potential infection, the user could be reidentified in order close contacts can be alerted for potential transmission of COVID-19 [3].

### 4.3. Societal and Business Aspects of COVID-19 LBMAs

As governments and stakeholders have been relying on digital technologies to address this novel threat, COVID-19 LBMAs increase the fight against COVID-19. To this end, we examine in-depth social and business implications to examine if these apps are providing solutions and are in line with expectations of the stakeholders and society.

Societal and business aspects in our analysis are related to the influence of these LBMAs on citizens in terms of user acceptance, governance model, and target users. We examined if some LBMAs try to provide more than a single functionality (multipurpose apps).

These criteria were selected as representative of their impact on citizens and business aspects and the availability of data. They should be taken into consideration to avoid discrimination and threats from the use of these apps and ease possible consideration of citizens regarding who and how is utilizing their sensitive data in commercial or non-commercial activities.

### 4.4. Legal Aspects of COVID-19 LBMAs: Privacy Concerns and Human Rights

Contact detection is a vital tool in the fight against COVID-19 that serves a short-term good for public safety, but it could be a long-term threat to the protection of our privacy. Smartphones can be used as a tool to identify an infected individual's contacts during an epidemic quickly. Nevertheless, it also raises serious privacy questions regarding the long-term disadvantages of increased surveillance activities, the custodian of the data, how it might be utilized, and how it should be governed.

Thus, on 20 March 2020, the Singaporean Ministry of Health released the "TraceTogether" (app no 123) for Android and iOS, which is designed to assist health officials in tracking down exposures after an infected individual is identified. The Trace Together app tracks via Bluetooth when two app users have been in close proximity. In Taiwan, authorities track phone location data for anyone under quarantine. "The goal is to stop people from running around and spreading the infection", said Jyan Hong-wei, head of Taiwan's Department of Cyber Security, who leads efforts to work with telecom carriers to combat the virus [33].

However, important privacy implications arise from the existence of such tracking apps, which have been used to expand mass surveillance, restrict individual freedoms,

and expose the most private details about individuals [19]. Privacy is a central feature of conversations related to mobile contact tracing apps, encouraging efforts to develop effective alternative solutions with more robust privacy protection for the users, while still achieving the public health goal of informing people of potential exposures to help slow the spread of the disease. A discussion regarding the implementation of app-based contact tracing to control the COVID-19 pandemic focuses on data protection and user acceptability aspects suggest the appropriate technology adoption. Such solutions will often increase the effectiveness and efficiency of the respective data-processing system. Only if people trust a system—because it does not spy on them—will the system find broad support in the population. Therefore, according to new technologies, quick and efficient contact detection is possible without extensive collection of amounts of data in a central database. Alternatives to the non-usage of location tracking and proximity tracing technologies have been implemented in Singapore which includes Bluetooth tokens that provide users with anonymity which might be an issue for many users [34].

Thus, with sufficient computational resources and the use of cryptographic protocols, app-based contact tracing can be accomplished without completely sacrificing privacy. We believe that developers and policymakers will carefully consider privacy issues when designing new contact tracing apps [35]. To this extent, decentralized protocols such as decentralized privacy-preserving proximity tracing (DP-3T) architecture [36] benefit the users with anonymity regarding their health status or their real-time location info. For the same reason, Bluetooth is the optimum solution to the proximity problem, which provides privacy preservation, something that GPS is not capable of doing up to now [3]. Most of the time, location data derived by GPS sensors are tied to known individuals (such as a name associated with a cell phone subscription), or it is tied to a unique identifier associated with a device. In these cases, individualized data is often referred to as "anonymized." In other cases, if a dataset has been modified to show movements of groups of people (and not individuals), which is often referred to as "aggregated" [37].

All the gathered data are overly sensitive, and so they are collected with higher security standards than usual. Mobile operators have higher responsibilities on this issue, and furthermore, they have established specific rules regarding COVID-19 LBMAs that they accommodate on both Android and iOS [38]. They can also evolve the existing LBMAs by providing advanced secured services during this pandemic that can help users stay safe [39]. Device manufacturers face new challenges for providing a safer digital environment with software updates that will allow mobile operators to provide enhanced privacy and services.

In general, the statements released by E.U. Supervisory Authorities so far suggest that the use of apps by public authorities to monitor the spread of COVID-19, will be allowed, provided that they comply with the principles contained in the laws governing EU data protection [2].

Governments across the world need to press pause on rolling out flawed or excessively intrusive contact tracing apps that fail to protect human rights. "If contact tracing apps are to play an influential part in combating COVID-19 people need to have confidence their privacy will be protected", said Claudio Guarnieri, Head of Amnesty International's Security Lab [40].

Operating LBMAs ought to seriously consider any potential limitations that may ensue due to the general data protection regulation (GDPR) and therefore readjust their terms and conditions of use. The use of LBMAs has raised new concerns about the trade-offs between health, citizens' wellbeing, and individual rights. Thus, researchers try to find a way to balance it all, reinventing the way of collecting and sharing personal data using new technological achievements to protect individual privacy [41].

## 5. Results

This section is divided into subheadings. We present our research results based on the order of our research framework, which is service aspects, technology aspects, societal and Business aspects, and legal aspects.

### 5.1. Service Aspects

Based on our framework we propose a categorization of COVID-19 LBMAs based on their service aspects. The results include the identification and inclusion of 160 LBMAs related to the COVID-19 with additional information in the Appendix of the current research. Our categorization on Table 1 includes: 79 "Health Status", 46 "Preserving Contact Tracing", 23 "COVID-19 Spread Mapping", 4 "COVID-19 Geosocial Media" and 8 multipurpose COVID-19 LBMAs. From these, 8 apps belong to the multipurpose category, 5 belong to health status and preserving contact tracing, and 3 to health status and COVID-19 spread mapping.

**Table 1.** COVID-19 LBMAs services' categories.

| Service Aspects | Numbering | Sum |
|---|---|---|
| Health Status | 1–79 | 79 |
| Preserving contact tracing | 80–125 | 46 |
| COVID-19 spread mapping | 126–148 | 23 |
| COVID-19 geosocial media | 149–152 | 4 |
| Health status and preserving contact tracing | 153–157 | 5 |
| Health status and COVID-19 spread mapping | 158-160 | 3 |
| Sum | 160 | 160 |

### 5.2. Technology Aspects

We identify the utilized technologies in 73 out of 160 COVID-19 LBMAs, and we categorized them based on these technologies. The symbol "-" on the tables that follow the current research means the nonexistence of relevant apps. Based on the previous section, we include the technologies: GPS, Bluetooth, Cellular Network, and 3 multipurpose categories as shown in Table 2.

**Table 2.** Utilized Technologies.

| | GPS | Bluetooth | Cellular Network | GPS/Bluetooth | GPS/Cellular | GPS/Bluetooth/ Cellular Network | Sum |
|---|---|---|---|---|---|---|---|
| Health Status | 9 | - | 4 | 1 | 19 | - | 33 |
| Preserving Contact Tracing | 10 | 9 | 1 | 5 | 9 | 1 | 35 |
| COVID-19 Spread Mapping | - | - | - | - | 1 | - | 1 |
| COVID-19 Geosocial Media | 1 | - | - | 1 | - | - | 2 |
| Health Status and Preserving Contact Tracing | 1 | 2 | - | - | - | - | 3 |
| Health Status and COVID-19 Spread Mapping | 21 | 11 | 5 | 7 | 29 | 1 | 73 |

*5.3. Societal and Business Aspects*

In terms of governance, our analysis of COVID-19 LBMAs is related to different stakeholders and targeted users. In our research, we identify the following stakeholders related to the development, governance, and consumption of services: businesses/organizations, government/public authorities, and end-users. Numerous mobile applications are developed and governed by businesses/organizations. These apps are named "Business Governed Apps (BGA)". Many applications have also been developed under public authorities' governance in order to monitor and control COVID-19 spread and ensure the safety of citizens that remain in quarantine [42] without compromising users' sensitive data. This data (such as location and movements) is mainly gathered and utilized by the public authorities in order to track the spread of Coronavirus [43]. In our analysis, these apps are named "Public Governed Apps (PGA.)". Users have developed a few LBMAs to assist other users to combat COVID-19. There is no governance model on these LBMAs, and we name them "peer to peer apps (P2P) "as presented in Table 3. When referring to end-users, we refer to the ones that download and use the LBMAs. All LBMAs related to COVID-19 in our sample are free to download and end-users that download those applications permit the publishers-developers of the apps to utilize their data including their location [44].

**Table 3.** Targeting of LBMAs.

| Type of Application | Governance Model | | | |
|---|---|---|---|---|
| Mobile Service | BGA | PGA | Peer to Peer | Sum |
| Health Status | 37 | 41 | 1 | 79 |
| Preserving Contact Tracing | 16 | 23 | 7 | 46 |
| COVID-19 Spread Mapping | 18 | 3 | 2 | 23 |
| COVID-19 Geosocial Media | 2 | 2 | - | 4 |
| Health Status and Preserving Contact Tracing | 1 | 4 | - | 5 |
| Health Status and COVID-19 Spread Mapping | 3 | - | - | 3 |
| Sum | 77 | 73 | 10 | 160 |

In our sample, we describe an LMBA as business to consumers (B2C) when the app is developed and published by a business/organization targeting end-users (consumers of apps) [41]. Government to consumers (G2C) when the app is developed or published by government/public authorities targeting end-users (consumers of apps), business to business (B2B), when the apps are developed and published by a business/organization targeting other businesses/organizations [45]. Finally, consumer to consumer (C2C) is a category where individuals develop LBMAs in order to distribute them to other individuals-consumers and utilize them in a peer-to-peer format [33].

As evidenced from Table 4, although in the "Health Status category" the results are divided almost equally between B2C and G2C, in the "COVID-19 Spread Mapping" category are mainly apps driven by businesses/organizations and target end-users. In the "Preserving Contact Tracing" category are mainly apps driven by businesses/organizations and target end-users. A limited number of apps targets other business users, whereas a significant number of LBMAs have been developed by citizens willing to contribute to the spread of COVID-19 actively.

**Table 4.** Targeting of LBMAs.

| Mobile Service | B2C | G2C | B2B | C2C | Sum |
|---|---|---|---|---|---|
| Health Status | 37 | 41 | - | 1 | 79 |
| Preserving Contact Tracing | 14 | 24 | 1 | 7 | 46 |
| COVID-19 Spread Mapping | 18 | 3 | - | 2 | 23 |
| COVID-19 Geosocial Media | 2 | 2 | - | - | 4 |
| Health Status and Preserving Contact Tracing | 1 | 4 | - | - | 5 |
| Health Status and COVID-19 Spread Mapping | 3 | - | - | - | 3 |
| Sum | 75 | 74 | 1 | 10 | 160 |

*5.4. Legal Aspects*

In several cases, the usage of LBMAs was deemed as mandatory (such as the case of "COVID19 Regione Sardegna"—app no. 31). In contrast, in most cases, end-users could voluntarily choose to utilize them. Some of the mobile applications that we have identified in our sample were evidenced as mandatory to use in specific countries, where public authorities forced the users to use them for any public transportation [46]. The majority of these apps were initially developed by private organizations and were adopted by governments to monitor users' location [45]. The governance of these apps is public, as governments forced the usage of the apps along with data utilization. The ones identified in our sample are presented in Table 5 below.

**Table 5.** Mandatory COVID-19 apps.

| Mobile Apps | Mobile Service | M-Commerce Model | Creator/Manager | Country | Technology | User |
|---|---|---|---|---|---|---|
| 4 | Health Status | G2C | Government and Alibaba | China-Hangzhou | GPS | Patients and non-patients |
| 31 | Health Status | G2C | Autonomous Region of Sardinia | Region of Sardinia | - | Patients and non-patients |
| 65 | Health Status | G2C | Ministry of Health and Welfare | South Korea | - | Patients and non-patients |
| 116 | Preserving Contact Tracing | G2C | Local Start-up | China-Hong Kong | GPS | Travelers |
| 153 | Health Status/ Preserving Contact Tracing | G2C | China Electronics Technology Group Corporation and government departments | China | - | Patients and non-patients |
| 154 | Health Status/Preserving Contact Tracing | G2C | TNeGA | India | GPS | Travelers |

## 6. Discussion

Most of the released COVID-19 LBMAs are Health Status. This might indicate that during the Coronavirus pandemic, the remote diagnosis was the most useful service regarding those apps. These numbers may also be increased since each country revealed its own Health Status app as an assistance tool for preliminary remote self-diagnosis of COVID-19. The Preserving Contact Tracing and COVID-19 Spread Mapping services might not have been developed for each country as the previous category but keep a high number as users need to trace possible infections near their environment.

Most COVID-19 LBMAs utilize location tracking technologies for their offering services in all four service categories of our study. Specifically, Health Status and Preserving Contact Tracing Service categories utilize GPS and Cellular Network mainly for their services. Proximity Tracing technologies are utilized mostly on the Preserving Contact Tracing

category where Bluetooth is utilized for the functionalities of these apps. Therefore, the technologies utilized in each app may be different depending on the service category that each app belongs.

For Societal and Business aspects an almost equal amount of LBMAs is governed by businesses/organizations and government/public authorities, indicating the significance of fighting COVID-19 by utilization of such means for both the private and public sector. Although the focus of BGAs is apart from Health Status mainly COVID-19 Spread Mapping, the focus of P.G.A.s is apart from Health Status, mainly Preserving Contact Tracing. To this end, it is essential to understand the potential in combining the efforts of both private and public organizations by providing pooled resources to develop reliable and targeted solutions.

We also witness a limited number of COVID-19 LBMAs targeting businesses. However, the need to develop better services that will allow them to normalize their business such as in-store contact tracing, support the Health Status of employees, and more, is essential to restore the economy to normality with safety.

Most of the mandatory COVID-19 LBMAs are based in Asia where the legal environment and the users' privacy protocols are different from the rest of the world. The strictest legal environment seems to be in Asia countries. Specifically, many of them originate from China where the rest of them from India, South Korea, and the Region of Sardinia. The remaining apps—shown in the Appendix—are being used voluntarily by mobile users worldwide.

## 7. Conclusions

A significant number of mobile applications released, rely on location information to improve the response to COVID-19. Different providers, involved stakeholders, and users struggle to define and understand the variety of ways that LBMAs can be applied within the value chain in epidemic management and monitoring, and in turn, conceptualize the provided services in ways that make sense for them. Despite the different perspectives and functionalities of the proposed four categories of LBMAs related to the purpose, value, and application, fundamental considerations and common questions exist across the development and commercialization phases of the various service types. These considerations include universal questions across the choice of provided functionalities/services, geographical market and service selection, target market and partnership, technical systems, and regulatory decisions.

The data provided by LBMAs can also be used to help healthcare professionals develop a better understanding of the virus and establish how science responds to it. The need for LBMAs related to COVID-19 may increase since restoring normal conditions will most probably be accompanied by a series of precautions that need to be taken to ensure the safety of citizens as well as the viability of commercial and public organizations. In most cases, the need for social distancing will still be in place, and one of the most convenient and safe ways to do so is to utilize existing mobile devices and technology. To this end, LBMAs should ensure that benefits will occur on personal, business, and governmental levels, while avoiding unwanted implications, such as privacy or safety infringements. We have identified a different number of involved entities that are influenced by COVID-19 LBMAs and might influence the launch of new LBMAs and the update of the existing ones.

COVID-19 LBMAs will continue assisting the global economy, especially at post lockdown phases that eventually many countries will face. Although the mandatory appliance of LBMA seems exaggerated in western countries, the voluntary character of such LBMAs seems to attract the majority as people tend to utilize them. Governments have the lead in the governance of launched applications. However, private business attempts for COVID-related apps can also keep a competitive level on the mobile apps' ecosystem and promote relevant synergies.

Mobile applications with different kinds of services have to adapt and provide information on how COVID-19 changes their offering services. Although LBMAs cannot trace

the number of people kept safe from getting infected in exact numbers, the estimation of those is something that might arrive as future research. Moreover, interest for future study, have the applications that were built on previously existing ones, to manage other similar crises or epidemics in the past. It seems that our community is negotiating new social contracts with mobile service providers and governments, and some of the results are already here as discussed in previous paragraphs. The solutions that have been developed from businesses and governments do not have to last forever, especially when they become mandatory. Still, they can become the starting point for building a new ecosystem for a future crisis response with respect to user privacy and rights.

This research comes with some limitations and urges for further research. The dataset of LBMAs and services is not conclusive. Although we gathered a significant number of applications through various sources, the landscape is changing every day. New apps are released while others are withdrawn. However, we tried to assess the landscape as a dynamic environment and point out the prominent trends as indicated in the time of this research and propose actions that will increase the utilization of LBMAs to increase their effectiveness in responding to the COVID-19 pandemic or future ones. Moreover, our analysis is based on primary results that appear to show significant trends. However, no in-depth analysis has been performed to outline the factors that will allow us to understand adoption patterns better. In this research, we focused on developing an initial framework of the released LBMAs and identifying the main factors that can lead to more efficient responses.

To this end, further research indicating the user's point of view through the utilization of technology acceptance models [47] and possible concerns they might have, will increase our knowledge on developing more effective and user-friendly LBMAs. Moreover, the in-depth analysis of the most downloaded applications can allow us to identify the main reasons for their adaptation, as a plethora of factors can lead to their success [48]. The way a service is designed has implications for the entire lifecycle of service development and will influence and define essential decisions that have to be made before taking it to market. Due to the fact that there is not yet a comprehensive view of the factors affecting the successful implementation of these applications in the lockdown era and the COVID-19 epidemic, a future area of research may be to identify the factors affecting the adoption of COVID-19 LBMAs regarding the perceived benefits and barriers that focus on users [49].

The success and effectiveness of the existing LBMAs in supporting governments, health authorities/World Health Organization, and citizens largely depends on the users' adaptability. The scalability of existing LBMAs can be enhanced by providing strong privacy guarantees using end-to-end encryption (E2EE), thus increasing the voluntary adoption of these applications [50].

**Author Contributions:** T.O., K.F. and M.V. equally contributed to this paper. All authors have read and agreed to the published version of the manuscript.

**Funding:** This research is co-financed by Greece and the European Union (European Social Fund-E.S.F.) through the Operational Programme «Human Resources Development, Education and Lifelong Learning» in the context of the project "Strengthening Human Resources Research Potential via Doctorate Research" (MIS-5000432), implemented by the State Scholarships Foundation (IKϒ).

**Institutional Review Board Statement:** Not applicable.

**Informed Consent Statement:** Not applicable.

**Data Availability Statement:** Data is contained within the article. The data presented in this study is available in the Appendix A.

**Conflicts of Interest:** The authors declare no conflict of interest.

# Appendix A

**Table A1.** List of COVID-19 Location-Based Mobile Applications utilized in the study (accessed March to June 2020).

| | No | Mobile Application | Links | Creator/Manager | Country |
|---|---|---|---|---|---|
| Health Status | 1 | Aarogya Setu | https://play.google.com/store/apps/details?id=nic.goi.aarogyasetu&hl=en&gl=US | NIC eGov Mobile Apps | India |
| | 2 | Ada—your health companion | https://play.google.com/store/apps/details?id=com.ada.app&hl=en&gl=US | Ada Health | Germany |
| | 3 | ADiLife Covid-19 | https://play.google.com/store/apps/details?id=it.adilife.covid19.app&hl=en&gl=US | ADiLife Srl | - |
| | 4 | Alipay Health code | https://www.csis.org/blogs/trustee-china-hand/chinas-novel-health-tracker-green-public-health-red-data-surveillance | Government & Alibaba | China-Hangzhou |
| | 5 | Asistencia COVID-19 GT | https://www.mspas.gob.gt/noticias/comunicados/content/11-coronavirus-covid-19.html | Red Ciudadana | - |
| | 6 | Babylon Health Services—Speak to a doctor, 24/7 | https://play.google.com/store/apps/details?id=com.babylon&hl=en&gl=US | Babylon Health | - |
| | 7 | Bihar COVID-19 | https://covidindia.org/bihar/ | hashTag | - |
| | 8 | Bingli | https://chat.mybingli.com/#/covid | - | - |
| | 9 | C Spire Health—UMMC Virtual COVID-19 Triage | https://play.google.com/store/apps/details?id=com.cspire.telehealth&hl=zh | C Spire | - |
| | 10 | Cachoeirinha Contra o Coronavírus | https://play.google.com/store/apps/details?id=br.com.sismu.covidmonitor&hl=pt | Sismu Software | - |
| | 11 | CG Covid-19 ePass | https://play.google.com/store/apps/details?id=com.allsoft.corona&hl=en&gl=US | AllSoft Consulting | India |
| | 12 | CoronApp—Colombia | https://play.google.com/store/apps/details?id=co.gov.ins.guardianes&hl=en&gl=US | INS.GOV | Colombia |
| | 13 | Coronavirus Australia | https://play.google.com/store/apps/details?id=au.gov.health.covid19&hl=en&gl=US | Department of Health, Australian Capital Territory | - |
| | 14 | Coronavirus UY | https://play.google.com/store/apps/details?id=uy.gub.salud.plancovid19uy&hl=en&gl=US | AGESIC | Uruguay |
| | 15 | COVA Punjab | https://play.google.com/store/apps/details?id=in.gov.punjab.cova&hl=en&gl=US | Government of Punjab | - |
| | 16 | COVI | https://play.google.com/store/apps/details?id=axiom.com.covi&hl=en&gl=US | Droobi Health Technology | Qatar |
| | 17 | Covi-ID | https://play.google.com/store/apps/details?id=com.coviid&hl=en&gl=US | Council for Scientific and Industrial Research, | South Africa |
| | 18 | Covid Symptom Tracker | https://www.clickorlando.com/health/2020/04/01/new-covid-19-symptom-tracker-app-helps-researchers-better-understand-coronavirus/ | Dr Tim Spector | - |
| | 19 | Covid-19 | https://www.adilife.net/en/covid-19/ | ADiLife Srl | - |
| | 20 | Covid-19 Armenia | https://play.google.com/store/apps/details?id=am.gov.covid19&hl=en&gl=US | EKENG CJSC | Armenia |
| | 21 | Covid-19 BA | https://covid-19.ba/ | Gobierno de la provincia de Buenos Aires | Boenos Aires |
| | 22 | Covid-19 Info | https://www.covid19-info.jp/covid-19-en.html | Hi-Solution | - |
| | 23 | Covid-19 Ministerio de Salud | https://www.argentina.gob.ar/aplicaciones/coronavirus | Government of Argentina | - |
| | 24 | COVID-19 NI | https://covid-19.hscni.net/ | Health & Social Care Northern Ireland | Northern Ireland |
| | 25 | COVID-19 screening tool | https://sharedhealthmb.ca/covid19/screening-tool/ | Apple/US Government | United States |
| | 26 | COVID-19 Screening Tool | https://www.apple.com/covid19 | Apple | - |
| | 27 | COVID-19 West Bengal Government | https://play.google.com/store/apps/details?id=com.pixxonai.covid19wb&hl=en&gl=US | SEMT Govt. of West Bengal | - |
| | 28 | COVID-19.eus | https://www.euskadi.eus/coronavirus-app-covid-eus/web01-a2korona/es/ | Osakidetza | Euskadi |
| | 29 | Covid-19MX | https://coronavirus.gob.mx/ | Zoe Global Limited | - |
| Health Status | 30 | COVID19 Paraná | http://www.coronavirus.pr.gov.br/Campanha/Pagina/Lancamento-aplicativo-COVID19-Parana | Celepar—Parana Information & Communication Technology Company | - |
| | 31 | COVID19 Regione Sardegna | https://www.appannie.com/en/apps/google-play/app/it.regione.sardegna.modulicovid19/ | AUTONOMOUS REGION OF SARDINIA | - |
| | 32 | Detector COVID-19 Toluca | https://play.google.com/store/apps/details?id=apps.toluca.detectorcovid_19&hl=gsw | Toluca City Council | Mexico |
| | 33 | Disinfection Checklist COVID19 | https://apps.apple.com/pa/app/disinfection-checklist-covid19/id1504450491 | Snappii | - |
| | 34 | Doctolib | https://play.google.com/store/apps/details?id=fr.doctolib.www&referrer=utm_source%3Dappbrain%26utm_medium%3Dappbrain_web%26utm_campaign%3Dappbrain_web | Doctolib | - |
| | 35 | Doctor's Manual—ICD-10, MES, SMP | https://gps-fake.com/spravochnik-vracha-mkb-10-mes-smp.html | LLC Medical Information Solutions | - |
| | 36 | eCOVID19 | https://www.malaysia.gov.my/portal/content/30953 | SCARLETRED Holding GmbH | Austria |
| | 37 | EDUS | https://edus.ro/ | Caja Costarricense de Seguro Social | Cost Rica |

**Table A1.** *Cont.*

| | No | Mobile Application | Links | Creator/Manager | Country |
|---|---|---|---|---|---|
| Health Status | 38 | FAMILY—COVID 19 | https://www.health.state.mn.us/diseases/coronavirus/prevention.html | Family Hospital | Vietnam |
| | 39 | GoK Direct—Kerala | https://play.google.com/store/apps/details?id=com.qkopy.prdkerala&hl=en&gl=US | Qkopy | India |
| | 40 | GVA Coronavirus | https://play.google.com/store/apps/details?id=es.gva.coronavirus&hl=en&gl=US | Valencian government | - |
| | 41 | Halodoc—Doctors, Medicine & Appointments | https://play.google.com/store/apps/details?id=com.linkdokter.halodoc.android&hl=en&gl=US | Halodoc | Indonesia |
| | 42 | HealthTap—24/7 Telemedicine | https://play.google.com/store/apps/details?id=com.healthtap.userhtexpress&hl=en&gl=US | HealthTap | - |
| | 43 | How We Feel app | https://consent.yahoo.com/v2/collectConsent?sessionId=3_cc-session_db0643bc-75f1-4434-a00c-269ec2877bba | Nonprofit organization | United States |
| | 44 | HSE COVID-19 | https://info.patientmpower.com/covid19-privacy-policy | patientMpower | Ireland |
| | 45 | HSE Video Consultation Service | https://www.webdoctor.ie/for-clinics | Webdoctor Limited | - |
| | 46 | Korona Önlem | https://play.google.com/store/apps/details?id=tr.gov.saglik.koronaonlem | Ministry of Health | Turkey |
| | 47 | Lynx-HCF Covid-19 | https://qode.healthcare/products-and-services/qode-lynx-hcf | QODE HEALTH SOLUTIONS | - |
| | 48 | Medscape | https://www.medscape.com/public/medscapeapp | WebMD, LLC | United States |
| | 49 | MeitY, Government of India | https://www.meity.gov.in/ | MeitY, Government of India | India |
| | 50 | MUSC COVID-19 Vital Link | https://play.google.com/store/apps/details?id=edu.musc.vitaltracker.defaultapp | Medical University of South Carolina | - |
| | 51 | MySejahtera | https://play.google.com/store/apps/details?id=my.gov.onegovappstore.mysejahtera&hl=en&gl=US | Malaysia Government | Malaysia |
| | 52 | NCOVI | https://play.google.com/store/apps/details?id=com.vnptit.innovation.ncovi&hl=en&gl=US | Ministry of Information and Communication | Vietnam |
| | 53 | Nepal COVID-19 Surveillance | https://play.google.com/store/apps/details?id=com.iclick.covidnew&hl=en&gl=US | iClick/Nepal Government | Nepal |
| | 54 | NICD COVID-19 Case Investigation | https://cci.nicd.ac.za/ | NICD COVID-19 Case Investigation | - |
| | 55 | Osiris HealthTech Systems | https://omi.app/covid-19 | - | - |
| | 56 | Pakistan's National Action Plan for COVID-19 | https://www.preventionweb.net/english/professional/policies/v.php?id=71206 | Aiwatech | Pakistan |
| | 57 | Parsek Information Technologies GmbH | https://parsek.com/information-supported-remote-collaboration | - | - |
| | 58 | PatientSphere for COVID19 | https://healthtransformer.co/new-app-tracks-covid-19-symptoms-mapping-hot-spots-for-public-health-officials-5767e2081282?gi=7a748b8e11c1 | Open Health Network | - |
| | 59 | Private Kit: Safe Paths | https://play.google.com/store/apps/details?id=edu.mit.privatekit&hl=en&gl=US | MIT | - |
| | 60 | PrivateTracer | https://www.privatetracer.org/ | Government | Netherlands |
| | 61 | Provincial Agency for Health Services—Autonomous Province of Trento | http://www.i-locate.eu/full_members/apss/ | Provincial Agency for Health Services—Autonomous Province of Trento | - |
| | 62 | Quarantine Watch | https://play.google.com/store/apps/details?id=com.bmc.qrtnwatch&hl=en&gl=US | Government of Karnataka | India |
| | 63 | Rakning C-19 | https://play.google.com/store/apps/details?id=is.landlaeknir.rakning&hl=en&gl=US | Medical Director of Health | Iceland |
| | 64 | Self Shield | https://sshield.org/ | Commonwealth Centre for Digital Health | Sri Lanka |
| | 65 | Self-diagnosis app | http://ncov.mohw.go.kr/selfcheck | Ministry of Health and Welfare | South Korea |
| | 66 | Self-Quarantine app | https://www.coviid.me/ | Council for Scientific and Industrial Research, University of Cape Town | South Korea |
| | 67 | SOS CORONAVIRUS | https://www.web24.news/u/2020/04/in-mali-an-application-to-monitor-the-coronavirus.html | AGETIC MALI | - |
| | 68 | Stop COVID-19 K | https://www.kg.undp.org/content/kyrgyzstan/en/home/blog/2020/how-does-digital-technology-help-crack-the-pandemic-in-kyrgyzsta.html | Center for Digital Technologies | Kyrgyz Republic |
| | 69 | STOP COVID19 CAT | https://play.google.com/store/apps/details?id=cat.gencat.mobi.StopCovid19Cat&hl=en_US&gl=US | Generalitat de Catalunya | Spain |
| | 70 | StopCovid | https://play.google.com/store/apps/details?id=fr.gouv.android.stopcovid&hl=en_ZA | Government of France | France |
| Health Status | 71 | StopTheSpread COVID-19 | https://play.google.com/store/apps/details?id=com.virustracker.app&hl=en&gl=US | Binary Mango | United Kindgom |
| | 72 | SwissCovid | https://github.com/DP-3T/dp3t-app-android-ch | Ubique, EPFL | Switzerland |
| | 73 | Tabaud (COVID-19 KSA) | https://play.google.com/store/apps/details?id=sa.gov.nic.tabaud&hl=en&gl=US | Saudi Data and Artificial Intelligence Authority (SDAIA) | Saudi Arabia |
| | 74 | Telemedicine of Ugra | https://uriit.ru/en/about/ | Medical Information Aanalytical Center, GBU | - |
| | 75 | Test Yourself Goa | https://play.google.com/store/apps/details?id=com.innovaccer.testyourselfgoa&hl=en&gl=US | Innovaccer Inc | United States |

**Table A1.** *Cont.*

| | No | Mobile Application | Links | Creator/Manager | Country |
|---|---|---|---|---|---|
| Preserving Contact Tracing | 76 | Test Yourself Puducherry | https://play.google.com/store/apps/details?id= com.innovaccer.testyourselfpuddu&hl=en_IN | Innovaccer Inc | India |
| | 77 | ViruSafe | https://virusafe.info/ | Bulgarian Government | Bulgaria |
| | 78 | WebMD: Check Symptoms, Find Doctors, & Rx Savings | https://play.google.com/store/apps/details?id= com.webmd.android&hl=en&gl=US | WebMD, LLC | United States |
| | 79 | Who MyHealth | https://www.who.int/ | Former Google & Microsoft Employees | United States |
| | 80 | AMAN | https://play.google.com/store/apps/details?id=jo. gov.moh.aman&hl=en&gl=US | Al-Hassan Hleileh | Jordan |
| | 81 | Apturi Covid | https://play.google.com/store/apps/details?id=lv. spkc.gov.apturicovid&hl=en&gl=US | Ministry of Health | Latvia |
| | 82 | BeAware Bahrain | https://play.google.com/store/apps/details?id=bh. bahrain.corona.tracker | Information & eGovernment Authority | Bahrain |
| | 83 | coEpi | https://www.coepi.org/ | individuals | United States |
| | 84 | Contact Tracer | https://contacttracer.ru/ | SoftTree | Russia |
| | 85 | Corona 100m | https://www.brusselstimes.com/opinion/108594 /corona-apps-south-korea-and-the-dark-side-of- digital-tracking/ | Lee Jun-young (Corona Map) | South Korea |
| | 86 | Corona Kavach | https://www.financialexpress.com/industry/ technology/government-of-indias-corona-kavach- covid-19-tracking-app-explained-in-10-simple- points/1913422/ | Ministry of Electronics & Information Technology | India |
| | 87 | Corona Watch | https: //www.thehindu.com/news/national/karnataka/ corona-watch-app-launched/article31193062.ece | KSRSAC KGIS | India |
| | 88 | Corona-Warn-App | https://play.google.com/store/apps/details?id=de. rki.coronawarnapp | individuals | Germany |
| | 89 | COVIDSafe | https://www.health.gov.au/resources/apps-and- tools/covidsafe-app | Australian Government Department of Health | Australia |
| | 90 | CovidSafe | https://covidsafe.cs.washington.edu/ | Microsoft volunteers, University of Washington | United States |
| | 91 | Covid Watch | https://www.covidwatch.org/faq | Covid Watch | United States |
| | 92 | COVID-19 Contact-Confirming Application | https://www.mhlw.go.jp/stf/seisakunitsuite/ bunya/cocoa_00138.html | Ministry of Health | Japan |
| | 93 | COVID-19 Quarantine Monitoring system | www.Deccanherald.com | Pixxon Ai Solutions Pvt Ltd. | India-Tamil Nadu |
| | 94 | Covid-19 Tracker | https://www.coronatracker.com/country/russia/ | Russian Government | Russia |
| | 95 | CovTracer | https://play.google.com/store/apps/details?id= edu.rise.ihnilatis | RISE Research Centre of Excellence | Cyprus |
| | 96 | CoWin-20 | https://government.economictimes.indiatimes. com/news/digital-india/covid-19-govt-plans-to- launch-cowin-20-app-to-curb-community- transmission/74827737 | Ministry of Electronics & Information Technology | India |
| | 97 | diAry "Digital Arianna" | https://play.google.com/store/apps/details?id=srl. digit.diary&hl=en_US | Alessandro Bogliolo | Italy |
| | 98 | DOCANDU Covid Checker | https://innov.afro.who.int/global-innovation/ docandu-covid-checker-1831 | Greek Government with Entrepreneurs | Greece |
| | 99 | eRouška | https://erouska.cz/ | Ministerstvo zdravotnictví České republiky | Russia |
| | 100 | Gerak Malaysia | https://www.gerakmalaysia.gov.my/ | Malaysia Government | Malaysia |
| | 101 | GH COVID-19 Tracker App | https://play.google.com/store/apps/details?id= com.moc.gh&hl=en&gl=US | Ministry of Communication | Ghana |
| | 102 | Hamagen | https://govextra.gov.il/ministry-of-health/ hamagen-app/download-en/ | Health Ministry | Israel |
| | 103 | Hioh | https://ohioh.de/ | HioH | Germany |
| | 104 | Immuni | www.immuni.italia.it | - | Italy |
| | 105 | ito | ito-app.org | Individualw and Companies | Germany |
| | 106 | Mahakavach | https://play.google.com/store/apps/details?id= com.mahakavach&hl=en&gl=US | Maharashtra State Innovation Society | India |
| Preserving Contact Tracing | 107 | MyTrace | mysejahtera.malaysia.gov.my/intro | Ministry of Science, Technology and Innovation | Malaysia |
| | 108 | NOVID | https://www.novid.org/ | individuals | United States |
| | 109 | NZ COVID Tracer | tracing.covid19.govt.nz | Ministry of Health | New Zealand |
| | 110 | pedulilindungi | https://play.google.com/store/apps/details?id= com.telkom.tracencare&hl=en&gl=US | Ministry of Com. & Infor. Technology | Indonesia |
| | 111 | Plan Jalisco Covid-19 | https://play.google.com/store/apps/details?id= com.covid19.cgig&hl=es_419 | Secretary of Finance—Government of the State of Jalisco | Mexico |
| | 112 | ProteGo Safe | https://github.com/ProteGO-safe | - | Poland |
| | 113 | Rakning C-19 | https://play.google.com/store/apps/details?id=is. landlaeknir.rakning&hl=en&gl=US | Government of Iceland | Iceland |
| | 114 | Smittestop | https://play.google.com/store/apps/details?id= com.netcompany.smittestop_exposure_notification& hl=en&gl=US | Sundheds- og Ældreministeriet | Denmark |
| | 115 | Smittestopp | https://helsenorge.no/smittestopp | Proprietary | Norway |
| | 116 | StayHomeSafe | https://play.google.com/store/apps/details?id= com.compathnion.equarantine&hl=en&gl=US | Local Start-up | China-Hong Kong |

**Table A1.** *Cont.*

| | No | Mobile Application | Links | Creator/Manager | Country |
|---|---|---|---|---|---|
| | 117 | Steerpath | https://steerpath.com/covid19 | - | - |
| | 118 | Stop Covid | https://www.novid20.org/en/projects | Government of France | Georgia |
| | 119 | StopKorona! | stop.koronavirus.gov.mk/en | Proprietary | North Macedonia |
| | 120 | The Spread Project | https://manoellemos.com/ | Manoel Lemos | Brazil |
| | 121 | ThreatWatcher Mobile | https://play.google.com/store/search?q=ThreatWatcher%20Mobile&c=apps&hl=en&gl=US | Alertus | United States |
| | 122 | TraceCovid | https://tracecovid.ae/ | Health Authority—Abu Dhabi | - |
| | 123 | TraceTogether | https://www.tracetogether.gov.sg/ | Developed by Government Technology Agency (GovTech) | Singapore |
| | 124 | VírusRadar | https://virusradar.hu/ | Technology Ministry | Hungary |
| | 125 | Wiqaytna (وقايتنا "Our prevention") | https://www.wiqaytna.ma/ | Ministry of Interior | Morocco |
| COVID-19 Spread Mapping | 126 | BC COVID-19 Support | https://www.trackcovid-19.org/ | Province of British Columbia, Canada | |
| | 127 | Castor Covid-19 | https://www.castoredc.com/mobile-app-privacy-policy/ | Healthy Ageing | - |
| | 128 | Corona Map | https://coronamap.sa/ | National Health Information Center | Saudi Arabia |
| | 129 | CoronaFACTS | https://apps.apple.com/nz/app/coronafacts/id1504490714 | Trusted Medical LLC | - |
| | 130 | Coronavirus Bolivia | https://www.boliviasegura.gob.bo/ | AGETIC BOLIVIA 2020 | - |
| | 131 | Coronavirus Pro | iphonehacks.com | - | - |
| | 132 | Covid Community Alert | https://www.trackcovid-19.org/ | Covid Community Altert | Italy |
| | 133 | COVID Racial Data Tracker | https://covidtracking.com/race | - | United States |
| | 134 | COVID Shield | https://www.covidshield.app/ | Volunteers | Canada |
| | 135 | COVID-19 | https://ncov.moh.gov.vn/web/guest/-/huong-dan-cai-app-suc-khoe-viet-nam | Electronic Health Administration | Vietnam |
| COVID-19 Spread Mapping | 136 | COVID-19 Daily | https://appadvice.com/game/app/covid-19-daily/1506427926 | Wrights Creative Services, L.L.C. | - |
| | 137 | COVID-19 GOV PK | https://play.google.com/store/apps/details?id=com.govpk.covid19 | National Information Technology Board | - |
| | 138 | COVID-19 Medischdossier | https://pharmaphorum.com/views-and-analysis/covid-19-puts-nhs-app-library-to-the-test/ | Uitgeverij The Optimist B.V. | - |
| | 139 | Covid-19 Tam | https://www.halifaxtoday.ca/coronavirus-covid-19-national-news/evolving-science-reason-for-inconsistent-messaging-on-covid-19-tam-2814479 | Gobierno Del Estado de Tam | - |
| | 140 | COVID-19!—The current spread of disease | https://play.google.com/store/apps/details?id=cz.nmbbrno.covid&hl=en_US&gl=US | Nemocnice Milosrdnych bratri, p.o. | - |
| | 141 | Covidom Patient | https://play.google.com/store/apps/details?id=fr.aphp.covidom&hl=el | Assostance Publique-Hopitaux de Paris | - |
| | 142 | Healthlynked COVID-19 Tracker | https://www.healthlynked.com/corona-virus-tracker/ | - | - |
| | 143 | Karantinas | https://koronastop.lrv.lt/ | COVID-19 simptomai/informacija | - |
| | 144 | MeMeTeo | https://play.google.com/store/apps/details?id=com.memeteo.weather&hl=en_US | - | - |
| | 145 | nCOV tracker | https://www.doh.gov.ph/2019-nCov/Updates | - | - |
| | 146 | Relief Central | https://relief.unboundmedicine.com/relief | Unbound Medicine | - |
| | 147 | SM-COVID-19 | https://www.smcovid19.org/ | - | Italy |
| | 148 | Trackcovid-19.org | https://www.trackcovid-19.org/ | Government of India | India |
| COVID-19 Geosocial Media | 149 | MyGov | https://my.gov.au/LoginServices/main/login?execution=e2s1 | MyGovIndia | India |
| | 150 | uLouder | https://ulouder.com/ | Turkish entrepreneurs | United States |
| | 151 | WeChat | https://play.google.com/store/apps/details?id=com.tencent.mm&hl=en&gl=US | Tencent Holdings | China |
| | 152 | Whistleblowing Information | https://iran-hrm.com/index.php/2020/04/04/iran-prosecutes-whistleblower-doctor-for-unraveling-state-cover-up/ | US Government | Iran |
| Health Status & Preserving Contact Tracing | 153 | Close Contact Detector | https://www.bbc.com/news/technology-51439401 | China Electronics Technology Group Corporation & government departments | China |
| | 154 | COVID-19 Quarantine Monitor Tamil Nadu | https://play.google.com/store/apps/details?id=com.pixxonai.covid19&hl=en&gl=US | TNeGA | India |
| | 155 | NHS App | https://www.covid19.nhs.uk/ | | United Kingdom |
| | 156 | SafePathsKE | https://www.pathcheck.org/ | | United States |
| | 157 | Stopp Corona | https://play.google.com/store/apps/details?id=at.roteskreuz.stopcorona&hl=en&gl=US | Austrian red cross | Austria |
| Health Status & Covid-19 Spread Mapping | 158 | Coronavirus-SUS | https://play.google.com/store/apps/details?id=br.gov.datasus.guardioes&hl=en&gl=US | Governo do Brasil | Brasil |
| | 159 | COVID Symptom Study | https://covid.joinzoe.com/ | Zoe Global Limited | United Kingdom |
| | 160 | Teleatendimento Saúde | https://apkpure.com/teleatendimento-sa%C3%BAde/br.inf.ids.teleatendimentosaude | Zoe Global Limited | - |

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
