# Peer review of "A Multidimensional Analysis of Released COVID-19 Location-Based Mobile Applications"

_futureinternet, doi:10.3390/fi13110268_

Round 1

Reviewer 1 Report

Corona Virus tracking and mapping is a key application for location-based mobile applications. The authors collected and categorized a large number of applications from a range of international contexts. This extensive, categorized and analyzed collection by itself is of great scientific value making the paper a valuable and significant scientific contribution.

Section 4.2.1
While it is correct that mobile network based location tracking and GPS were among the first localization technologies, the most widely used option today, as far as I know, is based on WiFi-IDs, e.g., on Android.

l.298 DP-1T -> DP3T?
l.301 "... which provides privacy preservation, something that GPS is not capable of doing up to now [1]." This is making a long discussion too short, I think. 

Reviewer 2 Report

I suggest some comments as follows: 

  1. The authors should add section 3 comparing mobile applications used in the same research area during COVID-19 or before.
  1. Figure 2: COVID-19 LBMAs selection is not clear. 
  1. The authors should add more discussion about mobile applications used for multidimensional analysis of location. 
  1. What is the future research you suggest?
